# VLA-ATTC: Adaptive Test-Time Compute for VLA Models with Relative Action Critic Model

**Wenhao Li**[1] **Xiu Su**[2] **Yichao Cao**[2] **Hongyan Xu**[2] **Xiaobo Xia**[3] **Shan You**[4] **Yi Chen**[5] **Chang Xu**[1]

## Abstract

Vision-Language-Action (VLA) models have demonstrated remarkable capabilities and generalization in embodied manipulation. However, their decision-making relies on a fast, instinctive process that lacks deliberation. This strategy often leads to suboptimal or catastrophic actions when facing complex or ambiguous scenarios that require greater consideration. In this paper, we introduce **VLA-ATTC**, a framework that endows VLA models with adaptive test-time compute (TTC). VLA-ATTC employs an uncertainty-based "cognitive clutch" to dynamically transition from reflexive execution to a TTC deliberation phase when necessary. During TTC phase, a novel **Relative Action Critic** (RAC) model identifies the optimal action from generated candidates via pairwise comparisons. This relative mechanism replaces unstable absolute value estimation, significantly simplifying the learning objective. Furthermore, we introduce an efficient sampling strategy to amortize computational costs and an automated data pipeline that curates preference pairs without manual annotation. On the LIBERO-LONG benchmark, VLA-ATTC reduces the failure rate of the SOTA model PI0.5 by over 50%.

## 1 Introduction

The advent of VLA models (Team et al., 2024; Black et al., 2024; Kim et al., 2024; Shukor et al., 2025; Kim et al., 2025; Li et al., 2024; Xu et al., 2025c;b; Yang et al., 2026; Li et al., 2026a;b) marks a significant milestone in Embodied AI (Liu et al., 2025; Paolo et al., 2024; Memarian & Doleck, 2024; Duan et al., 2022; Fung et al., 2025; Feng et al., 2025; Li et al., 2025c;d; Chen et al., 2026; Li et al.,

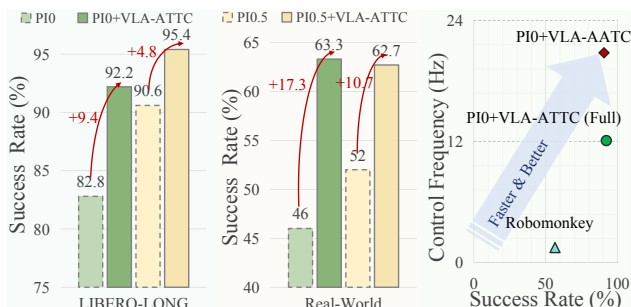

*Figure 1.* Performance overview and efficiency comparison.

2021; 2023; Tan et al., 2025). By leveraging the vast world knowledge embedded in pre-trained backbones (Wang et al., 2025a; Zeng et al., 2026; Li et al., 2025b), these models have demonstrated impressive generalization capability across diverse complex manipulation tasks. However, their decision-making is governed by a fast, intuitive inference process (Xu et al., 2025a). While this fast, intuitive strategy is sufficient for simple scenarios, it often leads to suboptimal or even catastrophic failures when facing complex or ambiguous situations that require deeper deliberation. This capability gap raises a fundamental question: *How can we endow VLA models with a powerful deliberation process, enabling them to evolve from fast, intuitive System 1 reasoning to the more considerate inference of System 2?*

Conceptually, deliberation can be classified into sequential and parallel paradigms (Muennighoff et al., 2025; Zhang et al., 2025; Li et al., 2025a; Huang et al., 2025; Zeng et al., 2025). Sequential approaches (Miao et al., 2024; Wang et al., 2025c), such as Chain-of-Thought (CoT) (Wang & Zhou, 2024; Xia et al., 2024; Chen et al., 2025; Yeo et al., 2025), have seen some exploration in VLA models (Zhao et al., 2025; Liu et al., 2024; Black et al., 2025; Zhou et al., 2025b;a; Song et al., 2025; Lin et al., 2025). However, these methods impose significant overhead, necessitating costly fine-tuning, laborious CoT data annotation, and often degrading action performance by forcing action-centric models to generate text reasoning.

In contrast, parallel deliberation strategies (Sessa et al., 2024; Zhang et al., 2024; Wang et al., 2025d; 2024b;a)—wherein the model generates a set of candidates and deliberates to select the one with the highest value—align better with the nature of embodied decision-

---

[1]University of Sydney [2]Central South University [3]University of Science and Technology of China [4]Sensetime Research [5]Hong Kong University of Science and Technology. Correspondence to: Xiu Su, Chang Xu <xiusu1994@csu.edu.cn, c.xu@sydney.edu.au>.

*Proceedings of the 43rd International Conference on Machine Learning*, Seoul, South Korea. PMLR 306, 2026. Copyright 2026 by the author(s).

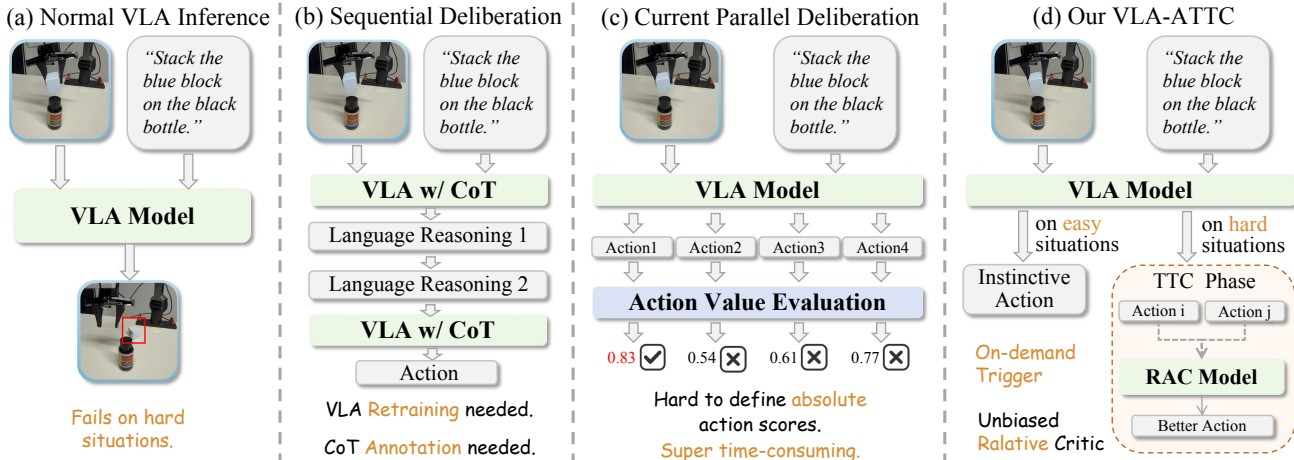

*Figure 2.* Comparison of VLA inference paradigms. (a) Standard Inference is fast but prone to failure in complex scenarios. (b) Sequential Deliberation needs costly retraining and data annotation. (c) Existing Parallel Deliberation incurs high costs and relies on unstable absolute scoring. (d) VLA-ATTC triggers deliberation only on hard situations, employing RAC model for robust candidate selection.

making. However, its exploration in VLA models remains nascent. Some works attempt (Kwok et al., 2025; Nakamoto et al., 2024) to employ a large-scale external critic model to select the best action from large number of candidates, while others (Guo et al., 2025) utilize a world model to guide exploration in a Monte Carlo Tree Search-like pattern. Although pioneering, these approaches apply deliberation to all scenarios indiscriminately and their deliberation processes are prohibitively costly, rendering them impractical for real-world application. This highlights that practical parallel deliberation methods for VLA models must address three critical challenges: *i) quantifying situational difficulty for adaptive deliberation*; *ii) minimizing the computational overhead of multi-action sampling*; and *iii) designing a lightweight yet high-fidelity action critic model.*

To address these challenges, we introduce VLA-ATTC, a framework that equips VLA models with adaptive test-time deliberation without modifying the base model. VLA-ATTC incorporates a "cognitive clutch" that monitors uncertainty in the base VLA's generation, triggering TTC deliberation phase only when necessary. During TTC phase, we mitigate candidates sampling costs via an efficient parallel strategy that amortizes vision-language computation. To solve the evaluation bottleneck, we propose a lightweight yet highly accurate relative critic model. This model shifts from predicting an absolute score to making a relative comparison—asking "Is action A preferable to action B?"—dramatically reduces the training difficulty and the required capacity of the evaluation model. Finally, to ensure scalability, we introduce an automated pipeline that curates high-quality preference pairs from existing manipulation datasets, eliminating the need for manual annotation. To sum up, our contributions are as follows:

- We propose VLA-ATTC, a plug-and-play framework that enables VLA models to adaptively trigger an efficient test-time deliberation phase in uncertain scenar-

ios, requiring no fine-tuning of the base model.
- We design a lightweight yet robust RAC model, which identifies optimal actions via iterative pairwise comparisons, overcoming the accuracy and efficiency bottlenecks of prior parallel deliberation methods.
- We develop a fully automated data pipeline that curates high-quality preference pairs directly from existing datasets, bypassing laborious manual data collection.
- Extensive experiments demonstrate that VLA-ATTC reduces LIBERO-LONG's SOTA failure rate by over 50% and boosts real-world success by 17.3% while maintaining a practical 20.8 Hz control frequency.

**Conflict of Interest Disclosure** None.

## 2 Related Work

**Sequential Deliberation in VLA models.** Early approaches like ECoT (Zawalski et al., 2024), CoT-VLA (Zhao et al., 2025), RoboMamba (Liu et al., 2024) and PI0.5 (Black et al., 2025) output predefined, structured information, such as sub-task decompositions, before an action. Subsequent works like ChatVLA (Zhou et al., 2025b), ChatVLA2 (Zhou et al., 2025a), Hume (Song et al., 2025), and OneTwoVLA (Lin et al., 2025) moved towards generating more adaptive, free-form natural language thoughts. However, all approaches demand costly fine-tuning to adapt the VLA for the auxiliary task of reasoning. Furthermore, they possess a conflict between the optimization of text and action.

**Parallel Deliberation in VLA models.** Parallel deliberation typically employs Best-of-N sampling (Sessa et al., 2024; Zhang et al., 2024; Wang et al., 2025d; Verdun et al., 2025; Toshniwal et al., 2025; Nguyen et al., 2025) or Self-Consistency (Wang et al., 2024b;a; Prasad et al., 2024; Wang et al., 2025b). For VLAs, one direction (Guo et al., 2025) uses a world model to guide Monte Carlo Tree Search, but is hampered by the immense computational cost and the lim-

ited fidelity of world models in real-world scenarios. Others (Kwok et al., 2025; Nakamoto et al., 2024) sample numerous action candidates and use a large external critic model for selection. However, their indiscriminate deliberation is costly, and the absolute action scoring is unstable.

## 3  Preliminaries: VLA Inference Paradigm

Contemporary VLA models, such as PI0 (Black et al., 2025), typically operate in a two-stage inference process. Given a multi-modal observation at timestep $t$, comprising an image $I_t$ and a language instruction $T$, the process is as follows:

**Vision-Language Encoding:** A high-capacity, pre-trained VLM backbone, $\Phi_{VLM}$, processes the inputs to produce a rich, multi-modal context embedding or inner features, $C_t$:

$$C_t = \Phi_{VLM}(I_t, T) \in \mathbb{R}^{L \times D_{\text{emb}}} \tag{1}$$

This context $C_t$ encapsulates the semantic understanding of the scene and the task objective. This operation, often involving a forward pass through a large Transformer, is computationally intensive.

**Action Decoding:** A specialized action head, $\Psi_{Action}$, then conditions on this context $C_t$ to generate an action chunk $a_t$:

$$a_t = \Psi_{Action}(C_t, z), \quad \text{where } \mathbf{z} \sim \mathcal{N}(\mathbf{0}, \mathbf{I}) \tag{2}$$

where $\mathbf{z}$ represents the initial noise vector or random seed required for the stochastic generation process. Modern VLAs predominantly employ generative models like diffusion or flow-matching for this stage.

Critically, the computational cost is highly asymmetric. The VLM encoding constitutes the vast majority of the latency, a process we term "pre-filling". In contrast, the action decoding, even with multiple sampling steps, is significantly faster. For example, in PI0 policy, action decoding accounts for only 27ms of the total 86ms inference time on RTX4090. Our VLA-ATTC framework is designed such that multiple action candidates can be sampled efficiently by performing the expensive pre-fill operation only once per timestep, amortizing its cost across all generated actions.

## 4  VLA-ATTC

### 4.1  Uncertainty Quantification as Clutch

A practical test-time deliberation method should only be activated when necessary to ensure manageable inference overhead. To evaluate the necessity of deliberation, we first need to quantify the model's uncertainty. Lower uncertainty implies greater model confidence, thus requiring less deliberation. We posit that a confident model will produce consistent actions despite variations in the random seed $z$, while an uncertain model will exhibit high variance.

At each timestep $t$, we generate two action chunks, $a_1$ and

---

**Algorithm 1** VLA-ATTC Inference

**Require:** VLM Backbone $\Phi_{VLM}$, Action Head $\Psi_{Action}$, Relative Action Critic $\mathcal{R}$, Image $I_t$, Instruction $T$, State $s_t$, Uncertainty threshold $\tau$, Candidate count $N$
1: $C_t \leftarrow \Phi_{VLM}(I_t, T)$       ▷ Prefill VLM once
2: $a_1, a_2 \leftarrow \Psi_{Action}(C_t)$     ▷ Generate two actions
3: $U_t \leftarrow \text{DTW}(a_1, a_2)$  ▷ Calculate DTW as uncertainty
4: **if** $U_t < \tau$ **then**             ▷ High confidence
5:     $a_{final} \leftarrow a_1$
6: **else**       ▷ Low confidence: trigger deliberation
7:     $\mathcal{A} \leftarrow \{a_i\}_{i=1}^N \leftarrow \Psi_{Action}(C_t)$ ▷ Batch Generation
8:     **while** $\text{LEN}(\mathcal{A}) > 1$ **do**
9:         $\mathcal{A}_{next} \leftarrow \emptyset$   ▷ Initialize next round's winners
10:         **for** $(a_i, a_j)$ in $\text{PAIRWISE}(\mathcal{A})$ **do**
11:             $p_{ij} \leftarrow \mathcal{R}(a_i, a_j, C_t, s_t)$    ▷ RAC predicts
12:             **if** $p_{ij} \geq 0.5$ **then**
13:                 $\text{ADD}(\mathcal{A}_{next}, a_i)$
14:             **else**
15:                 $\text{ADD}(\mathcal{A}_{next}, a_j)$
16:             **end if**
17:         **end for**
18:         $\mathcal{A} \leftarrow \mathcal{A}_{next}$       ▷ Update candidate set
19:     **end while**
20:     $a_{final} \leftarrow \mathcal{A}[0]$       ▷ Single best action remains
21: **end if**
22: $\text{EXECUTE}(a_{final})$

---

$a_2$, by sharing the VLM context $C_t$ but using different seeds:

$$\{\mathbf{a}_k\}_{k=1}^2 = \{\Psi_{\text{Action}}(\mathbf{C}_t, \mathbf{z}_k) \mid \mathbf{z}_k \overset{\text{iid}}{\sim} \mathcal{N}(\mathbf{0}, \mathbf{I})\} \tag{3}$$

The additional overhead from the second action is negligible with shared pre-filling. Let each action chunk be a sequence of $H$ poses, $a = (p^1, \dots, p^H)$, where each pose $p^k \in \mathbb{R}^D$ and $D$ is the action dimension. To measure the consistency between these two action chunks, we employ Dynamic Time Warping (DTW) distance.

To compute the DTW distance, we construct an $H \times H$ cumulative cost matrix $\Gamma$. Each element $\Gamma(i, j)$ stores the minimum cumulative cost required to align the first $i$ poses of $a_1$ with the first $j$ poses of $a_2$. This matrix is computed iteratively using the following recurrence relation:

$$\Gamma(i, j) = d(p_1^i, p_2^j) + \min \begin{cases} \Gamma(i-1, j) \\ \Gamma(i, j-1) \\ \Gamma(i-1, j-1) \end{cases} \tag{4}$$

where $d(p_1^i, p_2^j)$ is a local cost measure, which we define as the Euclidean distance $||p_1^i - p_2^j||_2$ between the two poses. The matrix is initialized with $\Gamma(0, 0) = 0$ and $\Gamma(i, 0) = \Gamma(0, j) = \infty$ for $i, j > 0$.

The final scalar uncertainty score $U_t$ is the total cumulative cost found in the top-right corner of the matrix, representing

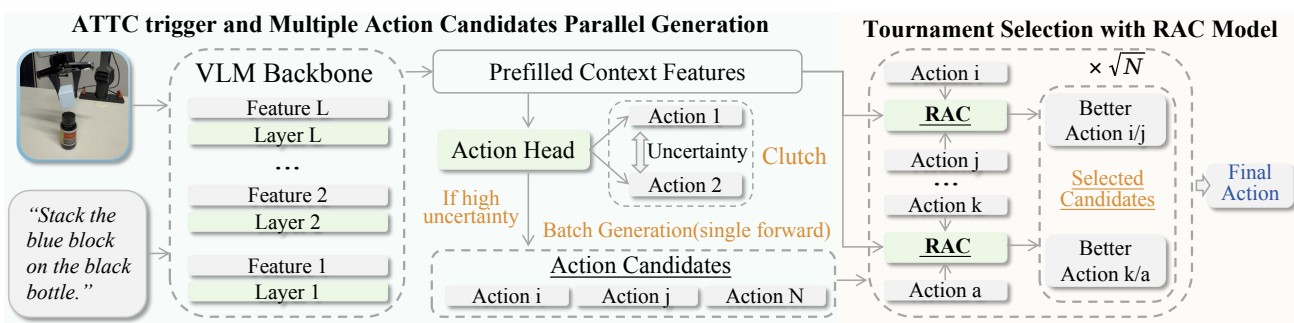

*Figure 3.* The overall architecture of the VLA-ATTC framework. The process begins with the "Cognitive Clutch" (Sec 4.1), which quantifies the VLA's action generation uncertainty. If uncertainty is low , the model executes the initial action reflexively. If uncertainty exceeds the threshold, the Deliberation Phase (Sec 4.2) is activated. This phase 1) efficiently generates N action candidates in parallel by amortizing the expensive VLM backbone computation, and 2) employs a Tournament Selection mechanism. Our lightweight Relative Action Critic model then performs iterative pairwise comparisons to identify the final optimal action.

the optimal alignment cost between the two full sequences:

$$U_t = \Gamma(H, H) \qquad (5)$$

The uncertainty score $U_t$ serves as a signal for our "Cognitive Clutch". We establish a threshold $\tau$ by taking the K-th percentile of uncertainty scores computed over an offline dataset. At test-time, the final policy $\pi_{\text{deploy}}$ adaptively switches between reflexive execution and TTC Deliberation:

$$\pi_{\text{deploy}}(\mathbf{s}_t) = \begin{cases} \mathbf{a}_1 & \text{if } \mathcal{U}_t \leq \tau \quad (\textit{Fast}) \\ \text{Deliberate}(\mathbf{C}_t, \mathbf{s}_t) & \text{if } \mathcal{U}_t > \tau \quad (\textit{Slow}) \end{cases} \qquad (6)$$

### 4.2 TTC Deliberation Phase

When TTC deliberation is triggered, we first generate $N$ candidate action chunks $\{a_1, ..., a_N\}$ parallel with the action head $\Psi_{Action}$ and prefilled context $C_t$:

$$\{a_1, ..., a_N\} = \Psi_{Action}(C_t, \{z_1, ..., z_N\}) \qquad (7)$$

The parallel batched sampling makes the cost of generating multiple candidates marginal.

Then we need to select the best action from these candidates. A significant challenge in this process is the inherent difficulty of action quality evaluation. Assigning an absolute, scalar quality score to a complex action chunk is often ill-defined and ambiguous. To circumvent this, our framework reframes the evaluation from absolute scoring to relative, pairwise comparison. We posit that determining whether action $a_i$ is preferable to $a_j$ (denoted $a_i \succ a_j$) given the current context is a more well-defined and less biased task. We leverage this principle by implementing a tournament-style selection process. This mechanism iteratively filters the candidates by conducting a series of pairwise comparisons. Let $\mathcal{R}$ be our RAC model, which is detailed in the following section. The RAC model can estimate the preference probability $p_{ij}$ given two actions, the proprioceptive state $s_t$, and the shared context $C_t$:

$$p_{ij} = \mathcal{R}(a_i, a_j, C_t, s_t) \qquad (8)$$

In each round of the tournament, the set of candidates is organized into pairs. For each pair $(a_i, a_j)$, the preferred action is selected to advance to the next round based on the critic's output:

$$a_{\text{winner}} = \begin{cases} a_i & \text{if } p_{ij} \geq 0.5 \\ a_j & \text{otherwise} \end{cases} \qquad (9)$$

This single-elimination process discards half of the candidates in each round. It repeats until a single, optimal action $a_*$ remains.

### 4.3 Relative Action Critic (RAC) Model

The cornerstone of our TTC deliberation phase is the RAC, a model trained to act as a pairwise preference critic. It determines if action $a_i$ is preferable to $a_j$ given the context. It is lightweight but powerful, as it can leverage the internal features of the VLM to reduce computation.

#### 4.3.1 Input Representation

The RAC takes four distinct inputs: two action chunks to be compared, $a_i$ and $a_j$; their difference $a_i - a_j$; and the current proprioceptive state $s_t$. Each input is independently mapped by a dedicated MLP into the model's embedding dimension $d_{model}$:

$$e_i = \text{MLP}_i(a_i), \quad e_j = \text{MLP}_j(a_j) \qquad (10)$$
$$e_{diff} = \text{MLP}_{diff}(a_i - a_j), \quad e_s = \text{MLP}_s(s_t) \qquad (11)$$

These four embedding vectors are then concatenated to form the initial hidden state for the RAC's Transformer tower: $X^0 = \text{Concat}[e_i; e_j; e_{diff}; e_s]$.

#### 4.3.2 Hierarchical Context Condition

To equip the RAC with task-relevant context, we introduce a set of $N_q$ randomly initialized, learnable query tokens $q_{rac} \in \mathbb{R}^{N_q \times d_{model}}$. These tokens are appended to the VLM's input sequence. During the VLM's prefilling computations, these queries distill high-level semantic information from VLM's raw features $F_{vlm}$, effectively creating

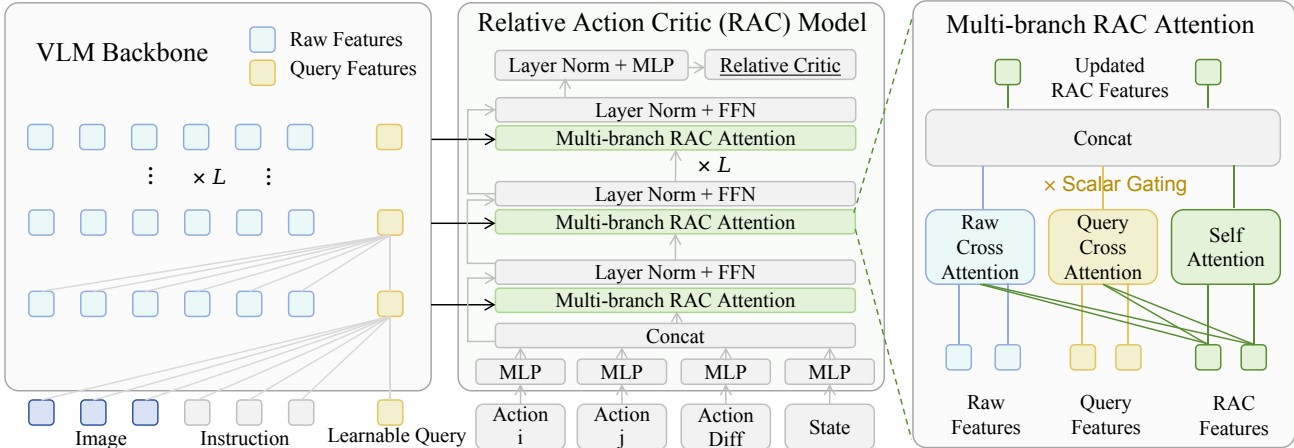

*Figure 4.* Detailed architecture of the Relative Action Critic (RAC) model (Sec 4.3). (Left) The VLM backbone is augmented with $N_q$ learnable queries. During the single pre-filling pass, these queries distill high-level semantic context query features from the image and instruction, alongside the standard raw features. (Middle) The RAC model itself is a Transformer with consistent depth of VLM. It takes four inputs processed by dedicated MLPs: Action i, Action j, their difference (Actioni - Action j), and the current proprioceptive State. (Right) The core component is the Multi-branch RAC Attention block. At each layer, this block hierarchically fuses information from three sources: (1) Self-Attention over the RAC's own features, (2) Raw Cross-Attention to the VLM's raw features at same layer, and (3) Query Cross-Attention to the distilled query features. This architecture allows the RAC to be lightweight yet highly context-aware.

a compressed, task-oriented summary of the VLM's understanding. The final hidden states of these query tokens, $F_{query}$, serve as a rich source of context for the RAC.

$$C_t = [F_{vlm}, F_{query}] = \Phi_{VLM}(I_t, T, q_{rac}) \quad (12)$$

#### 4.3.3 Multi-Branch Attention Architecture

The RAC is a Transformer of depth $L$, matching the VLM backbone's depth. Each layer $l \in [1, L]$ of the RAC updates its input $X^{l-1}$ through a sophisticated three-branch attention block, as depicted in Fig. 4.

1. **Self-Attention (SA):** The first branch computes self-attention over the RAC's own representations:

$$O_{sa}^l = \text{Attention}(Q(X^{l-1}), K(X^{l-1}), V(X^{l-1})) \quad (13)$$

2. **Raw Cross-Attention:** The second branch injects broad contextual information by attending to raw features of the corresponding layer $l$ in the VLM backbone, $F_{vlm}^l$:

$$O_{raw}^l = \text{Attention}(Q(X^{l-1}), K(F_{vlm}^l), V(F_{vlm}^l)) \quad (14)$$

3. **Query Cross-Attention:** The third branch attends to the specialized, distilled context features from the VLM's learnable query tokens from the same layer $l$, $F_{query}^l$:

$$O_{query}^l = \text{Attention}(Q(X^{l-1}), K(F_{query}^l), V(F_{query}^l)) \quad (15)$$

These three outputs are then fused. The query cross-attention branch is modulated by a learnable scalar gating parameter $g^l$, which allows the model to dynamically control the influence of the distilled context. The fused representation $O_{fused}^l$ is created by concatenation:

$$O_{fused}^l = \text{Concat}[O_{sa}^l, O_{raw}^l, g^l \times O_{query}^l] \quad (16)$$

This fused vector is then passed through the standard Transformer layer components: a feed-forward network (FFN) and layer normalization (LN), with a residual connection from the input $X^{l-1}$:

$$X^{l'} = \text{LN}(O_{fused}^l + X^{l-1}) \quad (17)$$

$$X^l = \text{LN}(\text{FFN}(X^{l'}) + X^{l'}) \quad (18)$$

After the final layer $L$, the output representation $X^L$ is processed by a classification head (an MLP) to produce a single logit. This logit is passed through a sigmoid function to predict the probability that $a_i$ is preferred over $a_j$, trained with a focal loss.

#### 4.4 Automated Action Preference Pair Curation

To train the RAC, we require a large-scale dataset of preference pairs. Thus, we introduce a fully automated pipeline that generates this dataset by manipulating the generation process of flow-matching-based action head.

**Principle of Conditional Flow-Matching.** The flow-matching action head learns a context-conditioned vector field that transforms a simple prior distribution $p_{prior}(a)$ (e.g., Gaussian noise) into the complex data distribution $p_{data}(a|C_t)$ of expert actions. Generation is achieved by solving the corresponding ODE numerically over a flow timestep $\tau$ from $\tau = 1$ down to $\tau = 0$:

$$\frac{da_\tau}{d\tau} = v(a_\tau, \tau, C_t) \quad (19)$$

The quality of the final sample is highly dependent on the number of integration steps ($N_{steps}$) used by the solver. A higher $N_{steps}$ yields a more accurate approximation of the

Table 1. Average task success rates (%) on the LIBERO-LONG. Ours (Full) represents VLA-ATTC without Cognitive Clutch.

| Task | Robomonkey | PI0 | +Ours (Full) | +Ours | PI0.5 | +Ours (Full) | +Ours |
|---|---|---|---|---|---|---|---|
| Soup and sauce in basket | 59% | 86% | 94% (+8%) | 92% (+6%) | 92% | 96% (+4%) | 96% (+4%) |
| Box and butter in basket | 79% | 98% | 100% (+2%) | 100% (+2%) | 96% | 100% (+4%) | 100% (+4%) |
| Turn on stove and put pot | 58% | 90% | 96% (+6%) | 94% (+4%) | 92% | 98% (+6%) | 96% (+4%) |
| Bowl in drawer and close | 37% | 96% | 100% (+4%) | 98% (+2%) | 98% | 98% (=) | 98% (=) |
| Two mugs on two plates | 55% | 90% | 98% (+8%) | 96% (+6%) | 94% | 100% (+6%) | 100% (+6%) |
| Book in compartment | 86% | 86% | 96% (+10%) | 94% (+8%) | 96% | 100% (+4%) | 98% (+2%) |
| Mug and pudding on plate | 59% | 84% | 94% (+10%) | 90% (+6%) | 90% | 98% (+8%) | 92% (+2%) |
| Soup and box in Basket | 62% | 96% | 98% (+2%) | 98% (+2%) | 98% | 98% (=) | 98% (=) |
| Both pots on stove | 26% | 40% | 58% (+18%) | 56% (+16%) | 54% | 68% (+14%) | 66% (+12%) |
| Mug in microwave and close | 44% | 62% | 88% (+26%) | 88% (+26%) | 92% | 98% (+6%) | 96% (+4%) |
| Average | 56.5% | 82.8% | 92.2% (+9.4%) | 90.6% (+7.8%) | 90.6% | 95.4% (+4.8%) | 94% (+3.4%) |

Table 2. Success rates (%) on real-world tasks with the Agilex Piper arm. VLA-ATTC consistently outperforms the baseline.

| Task | Robomonkey | PI0 | +Ours (Full) | +Ours | PI0.5 | +Ours (Full) | +Ours |
|---|---|---|---|---|---|---|---|
| Stack cubes | 18% | 46% | 62% (+16%) | 58% (12%) | 50% | 60% (+10%) | 60% (+10%) |
| Pour water | 24% | 50% | 66% (+16%) | 64% (+14%) | 54% | 68% (+14%) | 66% (+12%) |
| Sweep rubbish | 36% | 42% | 62% (+20%) | 56% (14%) | 52% | 60% (+8%) | 60% (+8%) |
| Average | 26% | 46% | 63.3% (+17.3%) | 58.7% (+12.7%) | 52% | 62.7% (+10.7%) | 62% (+10%) |

true trajectory along the vector field, resulting in a sample that is more faithful to the learned expert distribution.

**Data Curation Pipeline.** We leverage this principle to create actions of varying quality from a single pre-trained VLA. For each state-action pair $(o_t, a_t^{expert})$ from an expert dataset, we generate:

- **High-quality action** ($a_t^{good}$): By solving the ODE with a large number of steps, $N_{high}$.
- **Low-quality action** ($a_t^{bad}$): By solving the ODE with a small number of steps, $N_{low}$.

This establishes clear preference orderings: $a_t^{expert} \succ a_t^{bad}$ and $a_t^{good} \succ a_t^{bad}$, where $\succ$ denotes superiority. We then form a dataset of preference pairs with clear quality distinction, primarily using $\langle a_t^{expert}, a_t^{bad} \rangle$ and $\langle a_t^{good}, a_t^{bad} \rangle$. To ensure the model learns the relative quality rather than positional cues, we augment the dataset with the symmetric versions of each pair (e.g., also including $\langle a_t^{bad}, a_t^{expert} \rangle$ with the inverse label). This scalable, automated process allows us to generate a massive, high-quality preference dataset without any human intervention.

## 5 Experiments

In this section, we conduct extensive experiments to address the following research questions (RQ):

- **RQ1 (Effectiveness)**: Can VLA-ATTC significantly enhance the performance on complex tasks?
- **RQ2 (Ablation)**: What are the contributions of core components and the impact of candidate scaling and uncertainty threshold?
- **RQ3 (Mechanism)**: Are the proposed uncertainty measurement and automated data curation pipeline scientifically valid and necessary?

- **RQ4 (Efficiency)**: While delivering performance gains, does VLA-ATTC maintain the high control frequency required for practical applications?

**Environments and Tasks** Our experiments are conducted across a challenging simulation platforms LIBERO-LONG (Liu et al., 2023) and a real-world robotic system on an Agilex Piper Arm.

**Base Models and Implementation Details** We select PI0 (Black et al., 2024) and PI0.5 (Black et al., 2025), two SOTA VLA models, as our base models. Unless specified otherwise in our ablation studies, the uncertainty threshold $\tau$ is set to the 80th percentile, the number of learnable queries $N_q$ is set to 8, and the number of candidates is 16. The training step of RAC models in all experiments is 30000. For our automated data curation, we used a mixed set of two pairs for $(N_{high}, N_{low})$, specifically $\langle 10, 3 \rangle$ and $\langle 9, 2 \rangle$. "Ours (Full))" signifies that we do not use the "cognitive clutch", activating TTC at all time steps. Each task is executed 50 times to calculate the success rate.

### 5.1 Effectiveness of VLA-ATTC (RQ1)

**Obs1: VLA-ATTC boost the performance of SOTA VLA models significantly.** We first compare the performance of the original base VLA models against their VLA-ATTC-augmented counterparts on the LIBERO-LONG benchmarks and real-world settings. As shown in Table 1 and Table 2 , VLA-ATTC yields substantial performance improvements. On the highly challenging "Both pots on stove" task, VLA-ATTC improves the average success rate of PI0 from 40% to 58%, demonstrating that deliberate decision-making at critical moments effectively prevents catastrophic failures. Notably, VLA-ATTC increased the average success rate of PI0 by 17.3% on real-robot tasks, demonstrating its

*Table 3.* Effect of the uncertainty threshold ($\tau$, set by K-th percentile) on success rate and latency on PI0-ATTC and PI0.5-ATTC.

| K-th percentile | PI0-0% | PI0-40% | PI0-60% | PI0-80% | PI0.5-0% | PI0.5-40% | PI0.5-60% | PI0.5-80% |
|---|---|---|---|---|---|---|---|---|
| Soup and sauce in basket | 94% | 92% | 92% | 92% | 96% | 96% | 96% | 96% |
| Box and butter in basket | 100% | 100% | 100% | 100% | 100% | 100% | 100% | 100% |
| Turn on stove and put pot | 96% | 96% | 94% | 94% | 98% | 98% | 96% | 96% |
| Bowl in drawer and close | 100% | 98% | 98% | 98% | 98% | 98% | 98% | 98% |
| Two mugs on two plates | 98% | 98% | 98% | 96% | 100% | 100% | 100% | 100% |
| Book in compartment | 96% | 96% | 94% | 94% | 100% | 98% | 98% | 98% |
| Mug and pudding on plate | 94% | 94% | 92% | 90% | 98% | 96% | 96% | 92% |
| Soup and box in Basket | 98% | 98% | 98% | 98% | 98% | 98% | 98% | 98% |
| Both pots on stove | 58% | 58% | 58% | 56% | 68% | 66% | 66% | 66% |
| Mug in microwave and close | 88% | 88% | 88% | 88% | 98% | 96% | 96% | 96% |
| Average | 92.2% | 91.8% | 91.2% | 90.6% | 95.4% | 94.6% | 94.4% | 94% |

*Table 4.* Effect of the number of candidate actions ($N$) during deliberation on success rate on LIBERO-LONG.

| Number of candidate actions | 4 (PI0) | 8 (PI0) | 16 (PI0) | 32 (PI0) | 4 (PI0.5) | 8 (PI0.5) | 16 (PI0.5) | 32 (PI0.5) |
|---|---|---|---|---|---|---|---|---|
| Soup and sauce in basket | 90% | 90% | 92% | 92% | 94% | 96% | 96% | 96% |
| Box and butter in basket | 100% | 100% | 100% | 100% | 100% | 100% | 100% | 100% |
| Turn on stove and put pot | 92% | 94% | 94% | 94% | 94% | 96% | 96% | 98% |
| Bowl in drawer and close | 98% | 98% | 98% | 98% | 94% | 98% | 98% | 100% |
| Two mugs on two plates | 94% | 94% | 96% | 98% | 100% | 100% | 100% | 100% |
| Book in compartment | 90% | 94% | 94% | 96% | 98% | 100% | 98% | 100% |
| Mug and pudding on plate | 86% | 88% | 90% | 94% | 92% | 94% | 92% | 96% |
| Soup and box in Basket | 96% | 96% | 98% | 98% | 96% | 98% | 98% | 100% |
| Both pots on stove | 48% | 52% | 56% | 60% | 56% | 60% | 66% | 66% |
| Mug in microwave and close | 76% | 82% | 88% | 88% | 96% | 96% | 96% | 96% |
| Average | 87% | 88.8% | 90.6% | 91.8% | 92.0% | 93.8% | 94% | 95.2% |

excellent performance on physical hardware. VLA-ATTC consistently surpasses previous representative deliberation method Robomonkey (Kwok et al., 2025).

## 5.2 Ablation Studies and Analysis (RQ2)

**Obs2: Cognitive Clutch is accurate, with difficult states being sparse.** The sensitivity of the "Cognitive Clutch" is governed by the uncertainty threshold $\tau$, set as the K-th percentile. We study the effect of varying $\tau$ on task success rate. As shown in Table 3, although lower $\tau$ triggers more deliberation, the performance does not improve significantly. This suggests that only a few scenarios during the operation are difficult, which also proves that our cognitive clutch can effectively identify these few critical, difficult scenarios.

**Obs3: More candidate actions yield higher performance gains.** During the TTC deliberation phase, we explore better solutions by sampling $N$ candidate actions in parallel. We study how the choice of $N$ affects the final success rate. As presented in Table 4, performance improves substantially even with a small number of candidates (e.g., $N = 4$). While performance improves steadily as the number of candidate actions increases, we consider $N = 16$ to be an efficient and effective choice given the increased computational cost.

**Obs4: All specialized components of the RAC are essential for accurate preference prediction.** The ablation studies presented in Figure 5 show that removing key architectural elements—specifically the learnable weights, ac-

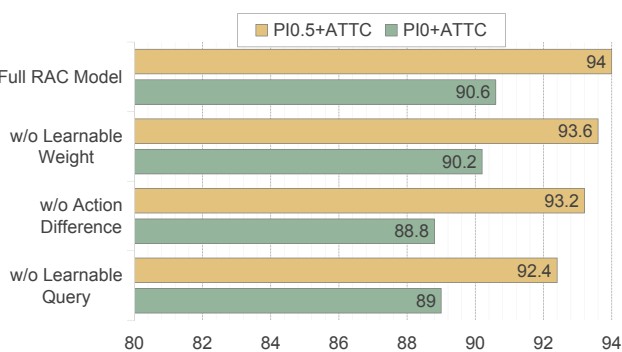

*Figure 5.* Comparison of success rate over different RAC model's variant on LIBERO-LONG.

tion difference inputs, or learnable queries—consistently degrades performance on the LIBERO-LONG benchmark. While the full RAC model achieves the highest success rates (94% for PI0.5+ATTC and 90.8% for PI0+ATTC), removing the "Learnable Query" component causes the most significant performance drop, lowering success rates to 92.4% and 88% respectively. These findings confirm that each design choice in the RAC framework is necessary to effectively evaluate and select optimal actions.

## 5.3 Mechanism Analysis (RQ3)

To validate the scientific validity of our core design choices—specifically the uncertainty measurement and the automated data curation pipeline—we conducted targeted controlled experiments.

*Table 5.* Comparison of average control frequency (Hz). VLA-ATTC maintains a high control rate.

| Model | PI0/0.5 (Baseline) | Robomonkey | VLA-ATTC (Full) | **VLA-ATTC** |
|---|---|---|---|---|
| Avg. Control Frequency (Hz) | 23.3Hz | 1.5Hz | 12.1Hz | 20.8Hz |

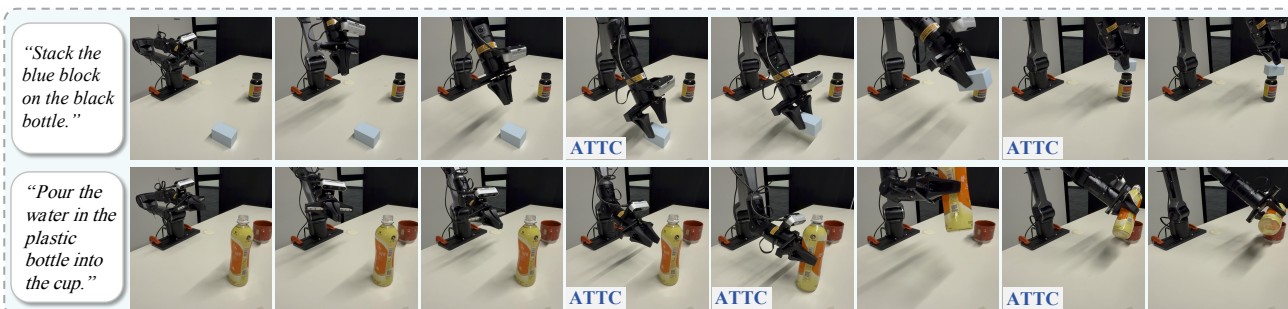

*Figure 6.* Two representative cases of PI0.5-ATTC. It adaptively triggers TTC when facing difficult situations.

*Table 6.* Validation of Uncertainty Estimation Strategies against Human Expert Rankings. Tested on 1,000 observation pairs.

| Method | Metric | Human Agree. (%) | Cost (Rel.) |
|---|---|---|---|
| Ours ($N = 2$) | Pairwise DTW | 89.2 | 1.0× |
| Baseline ($N = 4$) | Mean Pairwise Dist. | 90.1 | 2.0× |
| Baseline ($N = 8$) | Mean Pairwise Dist. | 90.4 | 4.0× |

*Table 7.* Impact of ODE Integration Steps on Action Quality. Real-world success rates on the "Pour water" task (50 trials each).

| ODE Steps ($N_{steps}$) | Success Rate (%) | Behavior Description |
|---|---|---|
| 10 (High Quality) | 54 | Precise execution |
| 5 (Mid Quality) | 42 | Coherent, misaligned |
| 1 (Low Quality) | 18 | Rough, frequent failure |

**Obs5: Minimal sampling ($N = 2$) is sufficient for effective uncertainty estimation.** A potential concern regarding our "Cognitive Clutch" is whether comparing merely two action candidates is robust enough to accurately gauge situational uncertainty. To investigate this, we constructed a specialized evaluation dataset comprising 1,000 pairs of observation scenarios. Four human experts annotated these pairs to identify which scenario in each pair presented a higher difficulty level (i.e., higher likelihood of failure), serving as the ground truth.

We compared our proposed strategy (2 actions with Dynamic Time Warping, DTW) against baselines using more samples ($N = 4$ and $N = 8$) utilizing Mean Pairwise Distance (MPD) as the uncertainty metric. As shown in Table 6, our method achieves an agreement accuracy of 89.2% with human experts. Crucially, scaling the sample size to 4 or 8 yields only marginal improvements ($< 1.5\%$) while linearly increasing the computational overhead of the action head.

**Obs6: Reducing ODE steps creates valid preference gradients, not random noise.** Our automated data curation pipeline relies on the premise that reducing the number of ODE integration steps ($N_{steps}$) degrades action quality gracefully without devolving into random noise, thereby creating valid "expert vs. sub-optimal" preference pairs. To validate this empirically, we conducted real-world experiments on the Agilex Piper arm using the "Pour water" task, varying $N_{steps}$ at 10, 5, and 1.

The results, presented in Table 7, confirm our hypothesis. While the high-quality setting ($N_{steps} = 10$) achieves a 54% success rate, reducing steps to 5 and 1 leads to a progressive decline in performance (42% and 18%, respectively). This reveals that actions generated with lower steps remain semantically coherent (e.g., moving generally towards the cup) but suffer from precision errors (e.g., spilling or misalignment), rather than exhibiting random jitter.

### 5.4 Efficiency Analysis (RQ4)

**Obs7: VLA-ATTC maintains high control frequency suitable for real-time robotics.** A core advantage of VLA-ATTC is its ability to improve decision quality while minimally impacting control frequency. Due to the "Cognitive Clutch", the expensive deliberation process is invoked only when necessary. As shown in Table 5, the baseline model operates at approximately 23.3 Hz. With VLA-ATTC integrated, the average control frequency drops only slightly to 20.8 Hz, as deliberation is active for only a few timesteps. This speed is significantly higher than other indiscriminate parallel deliberation methods that rely on heavyweight critic models or MCTS, such as RoboMonkey (Kwok et al., 2025) at 1.5 Hz. This demonstrates the immense practical advantage of our framework, making it suitable for robotic tasks that demand real-time responsiveness.

## 6 Conclusion

In this work, we introduced VLA-ATTC, a framework that fundamentally challenges the static, "one-size-fits-all" inference paradigm of VLA models. By dynamically triggering a test-time deliberation phase via an uncertainty-based "cognitive clutch", and employing an efficient, lightweight Relative Action Critic model for pairwise selection, our method significantly enhances decision-making robustness in complex scenarios without sacrificing the high control frequencies required for robotics. This work demonstrates the critical value of adaptive computation and opens a promising new direction for developing more efficient, deliberative, and intelligent agents that can strategically allocate

resources to the problem at hand.

## Impact Statement

This paper presents work whose goal is to advance the field of Machine Learning, specifically within Embodied AI and Robotic Manipulation. There are many potential societal consequences of our work, none which we feel must be specifically highlighted here.

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

# A    Verification of Semantic Preference Learning in RAC

A potential concern regarding our automated data curation pipeline (Sec. 3.4) is that the RAC model might solely rely on distinguishing low-level generation artifacts (e.g., trajectory smoothness or noise levels) resulting from different ODE integration steps ($N_{high}$ vs. $N_{low}$), rather than acquiring a semantic understanding of task fulfillment.

To rule out this possibility and verify that the RAC genuinely aligns actions with visual context and language instructions, we conducted a controlled "Cross-Task" experiment.

**Experimental Setup.** We utilized the real-world "Stack Cubes" task for this validation. We collected a dataset of 1,000 observations from successful execution trajectories. For each observation, we generated a pair of action candidates to be evaluated by the RAC model trained on the "Stack Cubes" task:

- **Positive Sample (Task-Aligned):** Actions generated by the "Stack Cubes" expert policy.
- **Negative Sample (Task-Misaligned):** Actions generated by a policy trained on a completely different task ("Pour Water"), applied to the "Stack Cubes" observations.

Crucially, **both policies generated actions using a high number of ODE integration steps** ($N_{steps} = 10$). This setup ensures that both candidates possess high trajectory quality and smoothness, eliminating low-level generation artifacts as a discriminative feature. The only distinguishing factor is the semantic alignment with the "Stack Cubes" task.

**Results.** The RAC model demonstrated a high accuracy of **97.3%** in identifying the actions from the "Stack Cubes" policy as the preferred candidates. This result strongly evidences that the RAC goes beyond detecting generation quality; it successfully learns to evaluate the semantic compatibility between the proposed action, the visual observation, and the task instruction.

