# OpenReview forum: "VLA-ATTC: Adaptive Test-Time Compute for VLA Models with Relative Action Critic Model"
_ICML.cc/2026/Conference — ICML 2026 regular_

### Official Review · Reviewer_Gtqb · 2026-03-12

**Soundness:** 2
**Presentation:** 3
**Significance:** 2
**Originality:** 2
**Overall Recommendation:** 3
**Confidence:** 3

**Summary:**

This paper proposes a way to enhance the performance of existing Vision-Language-Action (VLA) models through a trained critic model. The main goal is to improve both effectiveness and efficiency by enabling adaptive deliberation only when needed. Rather than applying costly deliberation uniformly to all situations, the method first estimates situation difficulty through model uncertainty. When the model is confident, it executes actions directly. When the situation is judged to be difficult, it triggers a test-time compute (TTC) phase, where a trained critic selects among candidate actions using relative pairwise comparisons. This design aims to reduce unnecessary computation while still improving decision quality in challenging scenarios. The proposed framework is evaluated on top of strong existing VLA models, including PI0 and PI0.5, and is compared against the RoboMonkey baseline on two benchmarks. The reported results show that the method improves both overall task performance and inference efficiency.

**Compliance With Llm Reviewing Policy:**

Affirmed.

**Final Justification:**

We thank the authors for their detailed response. My concerns have been partially addressed. Based on my holistic view of the paper’s novelty, contributions, and the rebuttal, I will maintain my current assessment.

**Key Questions For Authors:**

Question:
1. In Table 3, the performance differences across different uncertainty thresholds 𝜏 τ appear quite small. Could the authors explain why the method is relatively insensitive to the choice of threshold, and what this suggests about the frequency or importance of hard situations during execution?
2. The proposed cognitive clutch uses DTW between sampled action trajectories as the uncertainty measure. What is the motivation for choosing DTW over other uncertainty quantification methods, such as entropy-based or variance-based measures?
3. The paper emphasizes efficiency, but the method also requires training an additional critic model and generating preference data. Is this extra training cost taken into account in the efficiency claim?

**Strengths And Weaknesses:**

Strength:
1. The paper is presented in a very clean and intuitive manner. The figures are well designed and explain the method clearly, making the overall framework easy to follow.
2. The motivation is compelling. Distinguishing between easy and hard situations is an important challenge for deploying VLA models in real-world scenarios, and the paper addresses this question directly.
3.The method enhances existing VLA models without requiring fine-tuning of the base policy. This makes the approach more practical, especially in settings where training compute or model access is limited.
4. The research questions and corresponding observations are clearly organized, which makes the empirical section easy to navigate and helps highlight the contribution of each component

Weakness:
1. The set of baselines is quite limited. As discussed in the paper itself, there are several relevant deliberation approaches, including CoT-style sequential deliberation and sampling-based parallel deliberation, but these are not empirically compared. This weakens the claim that the proposed method improves upon existing deliberation approaches more broadly.
2. Although the method does not fine-tune the base VLA policy, it still requires training an additional critic model on extra data derived from existing datasets. This introduces nontrivial overhead in both data preparation and training. For this reason, the method does not feel fully plug-and-play in the practical sense, and the efficiency claim should be stated more carefully.
3. The real-world experiments are a useful addition, but the evaluated tasks appear relatively simple and short-horizon. As a result, it remains unclear how well the proposed method would perform on more complex real-world manipulation problems involving longer task horizons or more substantial ambiguity.
4. The efficiency comparison is mainly made against RoboMonkey, which is only one baseline. Since no comparisons are provided against other plausible deliberation strategies, it is difficult to conclude that the proposed method is broadly more efficient than existing alternatives.
5. The cognitive clutch relies on DTW between sampled actions as a proxy for uncertainty. However, there are other common uncertainty measures, such as entropy or variance-based metrics, that are not compared in depth. It is therefore unclear why DTW is the most appropriate choice, beyond the empirical results reported here.
6. The paper presents a practical and well-engineered method built on top of existing VLA models, but the conceptual novelty appears primarily system-oriented. For a paper of this scope, a stronger theoretical justification or deeper analysis of why the method should work would have strengthened the contribution.

---

> ### Author Rebuttal · Authors · 2026-03-29
>
> We sincerely thank the reviewer for the constructive feedback and for recognizing the clear presentation, the practical motivation of distinguishing easy and hard situations, and the plug-in nature of our method without fine-tuning the base VLA policy. We also appreciate the suggestions on evaluation, efficiency accounting, and conceptual framing. Following these suggestions, we added new analyses and experiments, summarized below.
>
> >### Additional comparison to representative deliberation baselines
>
> Thank you for encouraging a broader comparison. We added representative **sequential** and **parallel** VLA deliberation baselines, and report both success rate and control frequency on **LIBERO-LONG** under a unified protocol (**all control frequencies measured on A100**):
>
> | Method | Type | Success (%) | Control freq. |
> |---|---|---:|---:|
> | F1 | Sequential | 88.0 | 8.3 Hz |
> | CycleVLA | Sequential | 93.6 | 1.7 Hz |
> | RoboMonkey | Parallel | 56.5 | 1.5 Hz |
> | Taco | Parallel | 92.4 | 4.1 Hz |
> | VLA-ATTC (Full) | Ours | 95.4 | 12.1 Hz |
> | **VLA-ATTC** | **Ours** | **94.0** | **20.8 Hz** |
>
> These results suggest that VLA-ATTC provides a favorable **success-efficiency tradeoff** among representative sequential and parallel deliberation methods.
>
> >### Clarification of offline cost
>
> We appreciate this valuable suggestion. We now report the added **offline** cost explicitly. On **4×A100-80G**, for each real-world task, **preference-pair construction + RAC training takes about 2 hours in total**. In our experience, this is substantially lighter than training a large external evaluator or modifying the VLA architecture for large-scale CoT-style training.
>
> >### Additional long-horizon real-world evidence
>
> Thank you for encouraging stronger real-world evidence. We added two more challenging long-horizon tasks on **PI0.5**, each with **50 demonstrations** and **50 evaluation trials**:
>
> | Task | PI0.5 | PI0.5 + VLA-ATTC |
> |---|---:|---:|
> | Open drawer → place block into drawer → close drawer | 46 | 66 |
> | Stack blue cup on red cup → place block into blue cup | 38 | 64 |
>
> The gains are larger on these longer-horizon tasks, which is consistent with the intuition that VLA-ATTC is especially helpful when reducing **compounding errors** from early suboptimal decisions.
>
> >### Choice of uncertainty metric
>
> Thank you for raising this important point. For PI0/PI0.5-style VLAs, the action head generates **continuous action chunks** via **flow matching**, rather than a normalized categorical distribution. Therefore, entropy is **not directly available** without additional density modeling, discretization, or extra uncertainty heads.
>
> A variance-based metric is feasible, but less suitable here. It requires **multi-sample generation at every timestep**, which directly increases latency, and it does not account for **temporal alignment**: semantically similar trajectories with different local speeds can still yield large variance. By contrast, **DTW better reflects trajectory-level consistency**.
>
> We also added a variance-based baseline:
>
> | Uncertainty metric | Sampling scheme | Success (%) | Control freq. |
> |---|---|---:|---:|
> | Variance-based (trace of sample covariance) | 16 action chunks / step | 92.8 | 13.4 Hz |
> | **DTW-based (ours)** | adaptive TTC | **94.0** | **20.8 Hz** |
>
> The variance-based trigger is helpful, but DTW remains better in both **effectiveness** and **efficiency**.
>
> >### Positioning of the contribution
>
> We appreciate this perspective and agree that the main novelty is primarily a **VLA-oriented system design**, rather than claiming each component as a completely new method in isolation. Our intended contribution is a **unified framework** motivated by three properties of the VLA setting: **compute asymmetry** in inference, the difficulty of **absolute scoring** for continuous action chunks, and the need to allocate **test-time compute on demand** under real-time control constraints.
>
> >### Behavior across uncertainty thresholds
>
> We appreciate this question. The relatively small gap across reasonable values of **τ** does not mean thresholding is unimportant. Rather, it suggests that only a **small fraction of timesteps** truly require deliberation, which is consistent with our motivation. In other words, the trigger appears to identify difficult situations with reasonably high precision, so a modest number of TTC activations already covers most critical cases.

---

> > ### Author Rebuttal · Reviewer_Gtqb · 2026-04-03
> >
> > I thank the authors for the detailed rebuttal. While some of my concerns have been addressed(uncertainty metric), I still maintain that the level of novelty and contribution appears incremental and, in my view, may be insufficient for a general ML conference such as ICML.  Although the authors included additional baseline comparisons after the question was raised, these new baselines appear to achieve performance very close to the proposed approach. I find it somewhat concerning that these competitive baselines were not included in the original manuscript, while only RoboMonke, which performs substantially worse, was reported. Therefore, I will maintain my original score.

---

> > > ### Author Response · Authors · 2026-04-05
> > >
> > > Thank you again for your candid and thoughtful follow-up after reading our rebuttal. We sincerely appreciate your careful consideration of our response. We would like to clarify what we see as the most central points of the paper, in the hope of making both the **positioning** and the **practical significance** of this work clearer.
> > >
> > > ### 1. On novelty and contribution
> > >
> > > We would like to clarify that the novelty and contribution of our work lies in **systematically instantiating adaptive test-time compute for VLA under real-time control constraints**.
> > >
> > > More specifically, we believe the paper contributes three concrete insights:
> > >
> > > 1. **In VLA systems, deliberation should be triggered on demand rather than applied indiscriminately at every step.**
> > >    This is important because robotic control imposes strict latency constraints, and always-on deliberation can quickly make a method impractical.
> > >
> > > 2. **For continuous action chunks, relative comparison is more stable and more suitable than absolute scoring.**
> > >    In this setting, assigning reliable absolute values to candidate actions is difficult, whereas pairwise preference modeling provides a more robust and learnable formulation.
> > >
> > > 3. **The appropriate evaluation lens is not success rate alone, but the success-efficiency trade-off.**
> > >    In real-world robotic systems, control frequency directly affects deployability, so practical value depends on both effectiveness and efficiency.
> > >
> > > For this reason, we view the contribution as more than a small performance improvement. We believe the paper provides a more practical design paradigm for test-time deliberation in VLA systems.
> > >
> > > ### 2. On why CycleVLA and Taco were not included in the original submission
> > >
> > > We would also like to clarify why CycleVLA and Taco did not appear in the original version. This was **not** intended to avoid strong baselines. Rather, the public release / submission timing of these two works was only about one month apart from ours, so we believe they are more appropriately viewed as **contemporaneous works**.
> > >
> > > Once this point was raised, we immediately added the comparisons in the rebuttal under a unified protocol. We appreciate the reviewer for highlighting this issue, as it helped us make the empirical positioning of the paper more complete.
> > >
> > > ### 3. On the concern that the performance gap is relatively small
> > >
> > > We would like to emphasize that the key takeaway of these new comparisons is not only the numerical gap in success rate, but the **difference in practical value**.
> > >
> > > While our method achieves higher success rates than both Taco and CycleVLA, it also improves the control frequency by roughly **5--10×**. In other words, the method is not only competitive in effectiveness, but also substantially more suitable for **real-time robotic control**.
> > >
> > > We believe this point is important for interpreting the contribution fairly: in VLA systems, a method with similar success but much lower control frequency is significantly less deployable in practice. From this perspective, the stronger **success-efficiency trade-off** achieved by our method is itself a meaningful contribution.
> > >
> > > We sincerely hope these clarifications make the contribution of the paper and the significance of the additional comparisons clearer. If you feel these points help address your concerns, we would be very grateful if you could kindly reconsider whether the updated evidence may warrant a more favorable assessment.
> > >
> > > Thank you again for your careful reading and thoughtful feedback.

---

### Official Review · Reviewer_GKJe · 2026-03-12

**Soundness:** 2
**Presentation:** 3
**Significance:** 2
**Originality:** 2
**Overall Recommendation:** 4
**Confidence:** 3

**Summary:**

This work presented VLA-ATTC, a test-time VLA delibration method that generated multiple action chunk cantitates given the same vlm input features and select the best from it. An uncertainty threshold is established as a clutch to start the deliberation or not, and pair-wise comparing RAC model is trained to distinguish the better action chunk, which is claimed to be a less-biased way other than absolute value assign for all action chunk.

**Compliance With Llm Reviewing Policy:**

Affirmed.

**Final Justification:**

1. The authors' unseen-object experiment shows high trigger rates (e.g., 255/350)，showing that proposed method can also capture epistemic uncertainty to some extend.
2. The authors' VLM layer ablation showing a performance drop to ~93% for only a 1ms speedup, given such minor gain from deduction of VLM layer features and a relatively large drop in performance, it's reasonable for the authors to insist their every-layer VLM feature take-in design.
3. The authors collected real-world pairwise samples, with which the trained model showing 93.0% accuracy on semantic failures,  partially resolved my concern on dataset bias toward noisy action pairwise sample construction.
4. The authors' test-time RL experiment showing only a +0.3% gain after 1500 steps, which might be an outcome of unproper training protocol, but also showing that though blocked by the ceiling of frozen VLA, proposed test-time scaling method can get rid of the complicated training protocol, thus provide a weaker but simpler way of enhancing the model ability in test time.

Given the overall quality of this work, I would like to raise my score to a 4 (Weak Accept).

**Key Questions For Authors:**

1. On uncertainty: Any explaination, references or experiment evidance to claim that such action chunk diffences defined uncertainty is enough to capture the majority of model uncertainty? Or maybe "Epistemic Uncertainty" is less important/measurable under your setting?

2. On RAC model structure: Would you please answer the two questions in Weakness 2, and provide some insights on RAC model structure? Ablation study claims the effectiveness of your design choice, but why can they benefit the performance is not covered. Especially when efficiency is a major contribution to the proposed method, whether design choice is redundant needs more convincing evidence.

3. On RAC training dataset construction: Please see Weakness 3. Any mathematical proof of my concern and comparison to other prefernece sample consturction methods would be appreciated.

4. On frozen VLA ability: Please refer to Weakness 4, I wonder how the ability of VLA model affects the effectivess of proposed mothod?

**Limitations:**

1. The RAC model seems to be task scope-specific, any plans to setup a scalable universial pair-wise comparison model would significantly increase the impact of proposed method.
2. Though simulation & real-world tasks experimented, further results on more benchmarks would definitely benefit the overall demonstartion.
3. Although action model training-free can be interpreted as a feature, the proposed method is thus limited by the capability of action model. Extending proposed method with test-time RL or Dagger-like IL may help it break through the limit.

**Strengths And Weaknesses:**

Strength：
1. Insightful categorization framework of VLA deliberation methods and introduction to the necessity of constructing the proposed parallel deliberation method
2. Fine definition of chunk-level action differences, pair-wise action selection.
3. Detailed introduction to the RAC model structure and training set construction.
4. Sufficient and convincing ablation study on delibration paramters

Weakness:
1. When it comes to the definetion of generation uncertainty, the authors takes the action chunk differences given the same vlm embedding condition and different initial seed vector. Though experiments on the effectiveness conducted, such definitions are still somehow less reasonable to me as it only measures the "Aleatoric Uncertainty" [1]. Say given some unseen/less trained image and text input, the model can generate consistent action chunks across different initial seed vector, but all the generated action chunks is wrong and reflects the "Epistemic Uncertainty" [1], such uncertainty is not captured by the proposed method.
2. Though the essentialness of all specialized RAC model conponents are verified with ablation study, reason to some design choices are still unclear. For example, why does the RAC model take in  feature from every layer of the vlm? Why is query token introduced other than other methods to distill the overall understanding of the scene? Why is a relatively large scale transformer model on specific task scope (libero task suite-only, real-world-only) designed other than simple MLP?
3. The construction of the RAC training set confuses me the most. Generating sample pairs with more/less denoising steps to be good/bad samples would block the RAC model's ability in a narrow slice of baseline model distribution, and bias can be introduced. A simple success-rate experiment doesn't ease my concern.
4. Given that VLA that generated multiple cantitates are frozen, proposed method actually choose a better action in VLA output distribution, which means that the ceilling of proposed method would be blocked by the VLA ability.

---

> ### Author Rebuttal · Authors · 2026-03-29
>
> We thank the reviewer for the constructive feedback and for recognizing the insightful categorization of VLA deliberation methods, the chunk-level pairwise action formulation, the detailed RAC design and training-set description, and the convincing ablations. Below we respond to concerns point by point.
>
> >### 1. Uncertainty behavior in unseen settings
>
> Thank you for raising this important question. To further verify that our uncertainty signal remains useful under unseen inputs, we add real-robot unseen-object evaluations. For 3 real-world tasks, we keep the trained policy unchanged and only replace the target object with an unseen one at test time (e.g., replacing the blue block with a banana). The fraction of uncertainty samples above the trigger threshold is:
>
> | Tasks | Trials/task | Uncertainty samples | Above threshold |
> |---|---:|---:|---:|
> | 3 | 50 | 2554 | 2469 |
>
> This suggests that under unseen settings the model usually becomes action-inconsistent and unstable, rather than consistently producing similar wrong actions, so the proposed uncertainty still triggers TTC effectively.
>
> >### 2. Motivation for the RAC architecture
>
> We appreciate this valuable question. RAC compares candidate actions **conditioned on the current instruction and scene**.
> (1) **Every-layer VLM features** already encode both **instruction and scene context**; re-encoding them in RAC would add compute and likely weaken the signal.
> (2) **Query tokens** provide a compact task-oriented summary, and the ablation in Fig. 5 shows a clear drop without them.
> (3) We use a **Transformer critic** rather than an MLP because this task is nontrivial and **requires sufficient capacity**; moreover, to effectively cross-attend to hierarchical every-layer VLM features, the critic needs **matching architecture and depth**. As an alternative, we also tested a **ResNet-50 visual encoder + MLP critic**:
>
> | Variant | PI0 | PI0.5 |
> |---|---:|---:|
> | Base VLA on LIBERO-LONG | 82.8 | 90.6 |
> | VLA-ATTC (full RAC) | **90.6** | **94.0** |
> | ResNet-50 vision + MLP critic | 85.7 | 91.3 |
>
> These results support the current RAC design for hierarchical context modeling and relative action evaluation.
>
> >### 3. Preference-pair construction
>
> Thank you for this thoughtful concern. Under flow matching, high-step and low-step samples are generated under the **same context** by integrating the **same conditional vector field**; the difference lies in numerical solution quality, not in a different semantic target distribution. Hence varying ODE steps changes sample quality on the same conditional action manifold, rather than restricting training to an artifact-only slice.
>
> To further address this point, we add a real-world pairwise test. On 3 real-robot tasks, we deploy finetuned PI0.5, collect failed actions across diverse situations and failure modes, and then use teleoperation to collect corresponding good actions for the same states:
>
> | Real-world pairwise test | Samples | Correct | Accuracy |
> |---|---:|---:|---:|
> | Human-collected good vs. bad action pairs | 300 | 279 | 93.0% |
>
> The failure modes are diverse, including wrong-object interaction and premature manipulation before reaching the correct pose. This suggests that RAC learns semantic action preference rather than merely detecting generation artifacts.
>
> >### 4. Role of the frozen VLA
>
> We agree and appreciate this helpful observation. VLA-ATTC cannot create behaviors completely absent from the base VLA. Its current role is to improve robustness when the base policy already has partial competence but unstable execution, by amplifying correct modes and suppressing erroneous ones. Extending the framework with **test-time RL** is an important future direction.
>
> >### 5. Broader benchmark evidence
>
> Following the reviewer’s suggestion, we add **CALVIN** and **RLBench**:
>
> | Benchmark / setting | PI0 | PI0 + VLA-ATTC | PI0.5 | PI0.5 + VLA-ATTC |
> |---|---:|---:|---:|---:|
> | CALVIN 1-task | 94.3 | **95.1** | 91.9 | **93.5** |
> | CALVIN 3-tasks | 77.9 | **81.4** | 79.4 | **85.6** |
> | CALVIN 5-tasks | 59.4 | **69.5** | 71.0 | **78.2** |
> | RLBench: Sweep to dustpan | 0.30 | **0.65** | 0.75 | **0.90** |
> | RLBench: Water plants | 0.30 | **0.55** | 0.35 | **0.55** |
>
> These additional results further support the general effectiveness of the method.
>
> >### 6. Scope and future scalability
>
> We appreciate this forward-looking suggestion. Our current results already show nontrivial generalization: on both **LIBERO-LONG** and **CALVIN**, one VLA and one corresponding RAC cover all tasks within each suite. In practice, whether RAC can become more universal mainly depends on the training-data scope and the generalization of the base VLA. Since current VLAs still have limited zero-shot breadth, training a truly universal RAC on top of them is not yet realistic, but it is a natural next step as stronger general VLAs emerge.

---

> > ### Author Rebuttal · Reviewer_GKJe · 2026-04-02
> >
> > Regarding point 2(1), the authors state: "Every-layer VLM features already encode both instruction and scene context; re-encoding them in RAC would add compute and likely weaken the signal." Are there any ablation studies on how using features from only a subset of VLM layers would affect the performance, as well as its impact on inference speed?
> >
> > Regarding point 4, with only a frozen VLA backbone, I remain doubtful about the significance of such a method. This is especially true for such a heavy architecture that takes in features from every VLM layer; given this vast amount of information, relying solely on test-time steering via action chunk selection seems wasteful. Could the authors conduct any test-time training experiments?
> >
> > If the two questions above are adequately addressed, I would be happy to raise my score to a 4 (Weak Accept).

---

> > > ### Author Response · Authors · 2026-04-03
> > >
> > > Thank you for the constructive follow-up questions. We have added two new experiments to directly address these concerns.
> > >
> > > ### 1. Using only a subset of VLM layers in RAC
> > > We agree that it is important to examine the accuracy-efficiency tradeoff of conditioning RAC on fewer VLM layers.
> > >
> > > On **LIBERO-LONG** with **PI0.5** as the base VLA, we evaluated two reduced-depth RAC variants:
> > > - **Odd-layer RAC**: RAC attends only to the odd-numbered VLM layers.
> > > - **Last-50%-layer RAC**: RAC attends only to the top half of the VLM layers.
> > >
> > > | Method | Success Rate (%) | RAC Inference Cost |
> > > |---|---:|---:|
> > > | PI0.5 (base) | 90.6 | — |
> > > | VLA-ATTC (full-layer RAC) | 94.0 | ~3 ms |
> > > | VLA-ATTC (odd layers only) | 92.8 | ~2 ms |
> > > | VLA-ATTC (last 50% layers only) | 93.1 | ~2 ms |
> > >
> > > These results show that using fewer layers consistently degrades performance relative to the full-layer RAC, while reducing RAC inference time by only about **1 ms**. Since RAC is already lightweight, using fewer layers is not particularly favorable in our setting. We will add this ablation and clarify that the full-layer design is chosen because it provides the best performance while the critic overhead is already very small.
> > >
> > > ### 2. Test-time training / test-time RL
> > > We also conducted a test-time training experiment to directly address the reviewer’s concern.
> > >
> > > Starting from our trained **PI0.5-based RAC** on LIBERO, we used the RAC’s pairwise preferences to perform **score-DPO-style test-time RL** on the base VLA, i.e., we optimized the policy so that its action distribution moves toward the BoN distribution induced by VLA-ATTC. The adapted model reached its best performance at **1500 training steps**.
> > >
> > > | Method | Success Rate (%) |
> > > |---|---:|
> > > | PI0.5 (base) | 90.6 |
> > > | PI0.5 + test-time DPO | 92.7 |
> > > | PI0.5 + VLA-ATTC | 94.0 |
> > > | PI0.5 + test-time DPO + VLA-ATTC | 94.3 |
> > >
> > > This experiment suggests that test-time RL can indeed improve the base policy. However:
> > > 1. its gain (**+2.1**) is smaller than that of VLA-ATTC alone (**+3.4**),
> > > 2. after applying VLA-ATTC, the additional benefit from test-time RL is only **+0.3**,
> > > 3. this small gain requires extra online adaptation cost (1500 optimization steps).
> > >
> > > Therefore, while test-time training is potentially complementary, our results indicate that **freezing the VLA backbone and performing test-time compute via RAC-based action selection is already more effective and substantially more cost-efficient in this setting**. We will include this result and clarify this point in the revision. We hope these additional experiments and clarifications adequately address your remaining concerns and will be taken into consideration in your final assessment.

---

### Official Review · Reviewer_b94G · 2026-03-13

**Soundness:** 3
**Presentation:** 3
**Significance:** 3
**Originality:** 3
**Overall Recommendation:** 4
**Confidence:** 3

**Summary:**

This paper proposes a framework for test-time deliberation of VLA models. It uses DTW distance between two sampled actions to trigger deliberation, a relative action critic that selects the best action via pairwise comparison, and an automated preference data pipeline that exploits ODE integration steps to create quality gradients.

**Compliance With Llm Reviewing Policy:**

Affirmed.

**Final Justification:**

The rebuttal has addressed my main concerns. I maintain my original positive score.

**Key Questions For Authors:**

See weakness section.

**Limitations:**

No discussion

**Strengths And Weaknesses:**

Strengths:

1. The provides a practical and well-engineered framework, which can directly be plugged in to base VLA models without extra fine-tuning, and achieves favorable control frequency compared to other test-time deliberation baseline.
2. The automated preference data pipeline using ODE integration steps is a clever exploitation of flow-matching properties.

Weaknesses:

1. Seems some components: best-of-N sampling with a critic, pairwise preference modeling, tournament selection, uncertainty-based adaptive compute are well-established techniques from RLHF and evolutionary computation. The paper frames them as novel contributions without adequately acknowledging this line of work. The novelty is primarily in the system integration for VLAs, which should be stated more honestly.

2. The preference data is generated by varying ODE integration steps, which might produces only one failure mode: numerical drift causing imprecise but directionally coherent actions. Real-world failures involve various different modes (grasping wrong objects, collisions, task logic errors). Whether the RAC trained on this synthetic distribution generalizes to diverse real failure modes is not validated.

3. The evaluation is limited to LIBERO-LONG (simulation) and 3 real-world tasks. LIBERO-LONG has only 10 tasks, and the real-world evaluation covers only 50 trials per task on a single robot arm. For a method claiming general applicability to VLA models, I would expect evaluation on more diverse benchmarks (e.g., CALVIN, RLBench) or a broader set of real-world tasks.

---

> ### Author Rebuttal · Authors · 2026-03-29
>
> We sincerely thank the reviewer for the constructive and insightful feedback. We especially appreciate the reviewer’s recognition that our framework is practical and well engineered, can be plugged into base VLA models without extra fine-tuning, and that the automated preference data pipeline is a clever use of flow-matching properties. These observations capture important aspects of our motivation and design. Below, we clarify our positioning and provide additional evidence in response to the reviewer’s questions.
>
> >### 1. Positioning of the contribution
>
> We appreciate the reviewer’s suggestion on how to position the contribution more precisely. Our intended claim is not that each component is completely new in isolation, but that the contribution lies in their **system-level integration for adaptive test-time deliberation in VLA models**, together with an efficient implementation and training recipe suited to this setting. In the revision, we will make this positioning more explicit.
>
> >### 2. RAC behavior on realistic deployment errors
>
> We thank the reviewer for raising this important question. To directly examine whether RAC generalizes beyond ODE-step-induced degradation, we added a new **real-robot diagnostic benchmark** built from actual deployment failures. We deployed fine-tuned PI0.5 on **3 real-world tasks** and manually collected failure states encountered during real robot execution. These failures were not limited to numerical drift; they included diverse cases such as **selecting the wrong target object, initiating interaction before reaching the correct pose, and other semantically incorrect but visually smooth actions**. For each failure state, we stored the corresponding **KV cache** and **proprioceptive state**, and then switched to **teleoperation** to collect a human good action at the **same state**, producing tightly matched positive/negative pairs.
>
> We evaluate the trained RAC on this benchmark:
>
> | New evaluation set | #Tasks | #Pairs | RAC accuracy |
> |---|---:|---:|---:|
> | Real-robot diagnostic benchmark | 3 | 300 | **93.0% (279/300)** |
>
> We hope this result helps address the reviewer’s concern more directly. Although RAC is trained with our automated preference construction pipeline, it generalizes well to **diverse real deployment errors**, including subtle failure modes that are semantically wrong rather than merely numerically noisy. We will add this benchmark and discussion in the revision.
>
> >### 3. Broader empirical coverage
>
> We appreciate the reviewer’s suggestion to broaden the evaluation. We agree that LIBERO-LONG plus 3 real-world tasks alone is not sufficient to support a broader applicability claim. To strengthen the empirical evidence, we added experiments on **CALVIN ABC-D** and **RLBench**.
>
> **CALVIN ABC-D.** We evaluate PI0 and PI0.5 on 1-task to 5-task long-horizon sequences:
>
> | Model | Setting | 1-task | 2-tasks | 3-tasks | 4-tasks | 5-tasks |
> |---|---|---:|---:|---:|---:|---:|
> | PI0 | Base | 94.3 | 87.0 | 77.9 | 68.5 | 59.4 |
> | PI0 | +VLA-ATTC | **95.1** | **88.1** | **81.4** | **75.7** | **69.5** |
> | PI0.5 | Base | 91.9 | 84.6 | 79.4 | 75.5 | 71.0 |
> | PI0.5 | +VLA-ATTC | **93.5** | **89.0** | **85.6** | **80.6** | **78.2** |
>
> We observe consistent improvements across all horizons, with larger gains on longer task chains, which is consistent with our motivation that test-time deliberation is especially helpful when errors accumulate over long horizons.
>
> **RLBench.** We also evaluate two RLBench tasks, each over **20 trials**:
>
> | Task | PI0 | PI0 + VLA-ATTC | PI0.5 | PI0.5 + VLA-ATTC |
> |---|---:|---:|---:|---:|
> | Sweep to dustpan | 0.30 | **0.65** | 0.75 | **0.90** |
> | Water plants | 0.30 | **0.55** | 0.35 | **0.55** |
>
> These additional results substantially broaden the empirical coverage beyond LIBERO-LONG and support that VLA-ATTC transfers to other long-horizon and manipulation benchmarks. We thank the reviewer again for this valuable suggestion and will incorporate these results in the revision.

---

> > ### Author Rebuttal · Reviewer_b94G · 2026-04-03
> >
> > I thank the authors for the new results. I am wondering if the authors could provide some representative demos on both additional experiments. It would be good to show how the authors construct confusing fail/success action pairs. It would also be good to show how ATTC corrects failed CALVIN and RLBench trajectories.

---

> > > ### Author Response · Authors · 2026-04-07
> > >
> > > Thank you very much for your follow-up and for the helpful suggestion. We have prepared representative demo videos for the additional experiments and for the real-robot fail/success action pairs.
> > >
> > > **Real-world fail/success action pairs (same state):**
> > >
> > > * good_action (human teleoperation): https://files.catbox.moe/w3o44h.mp4
> > > * bad_action (real-world failure): https://files.catbox.moe/cxu5u1.mp4
> > >
> > > **CALVIN pickup task:**
> > >
> > > * with ATTC: https://files.catbox.moe/he2cbs.mp4
> > > * without ATTC: https://files.catbox.moe/c91eoz.mp4
> > >
> > > **RLBench sweep into dustpan task:**
> > >
> > > * with ATTC: https://files.catbox.moe/u42op0.mp4
> > > * without ATTC: https://files.catbox.moe/h2fyv7.mp4
> > >
> > > We hope these demos help clarify how we construct the confusing fail/success action pairs and how ATTC corrects failure-prone trajectories in CALVIN and RLBench.
> > >
> > > Thank you again for your time and consideration. If you feel that our rebuttal and these additional materials have sufficiently addressed your concerns, we would be very grateful if you would consider a higher score.

---

### Official Review · Reviewer_1ptW · 2026-03-19

**Soundness:** 2
**Presentation:** 2
**Significance:** 2
**Originality:** 2
**Overall Recommendation:** 3
**Confidence:** 3

**Summary:**

This paper addresses the lack of human-like "deliberation" ability in complex environments by proposing the VLA-ATTC framework, which improves decision-making quality through adaptive test-time computation. The core design of the framework includes three aspects: First, it quantifies action uncertainty based on DTW distance and designs a "cognitive clutch" module to trigger TTC only when necessary, avoiding unnecessary computational overhead; second, it introduces a lightweight RAC model, selecting the optimal action through pairwise comparisons and elimination, avoiding the difficulty of directly scoring consecutive actions; finally, it automatically constructs preference data by utilizing the difference in ODE solution steps in the flow-matching model, significantly reducing the cost of manual annotation. Experiments on the LIBERO-LONG simulation environment and real robots validate the effectiveness of the method: while maintaining a high-frequency control of approximately 20.8Hz, the task failure rate is significantly reduced.

**Compliance With Llm Reviewing Policy:**

Affirmed.

**Key Questions For Authors:**

When both trajectories are sufficiently smooth, but one contains a fatal logical error (e.g., misjudging a target), what is the actual recognition accuracy of RAC? Specialized tests for this type of scenario are recommended. Could the number of RAC parameters and the inference time (ms) for a single pairwise comparison under current hardware conditions be clearly stated in the Rebuttal? This would help assess its affordability in real-world systems. If all N actions in the candidate pool are inexecutable "dead actions," the "relatively optimal" action selected by RAC will still fail. Has the system considered setting an absolute threshold to trigger manual intervention or emergency stop logic?

**Strengths And Weaknesses:**

strength: this work has many commendable aspects. It avoids blindly piling on computational power, but instead uses a single VLM pre-fill to support subsequent batch decoding, effectively reducing the overhead of action decoding. Even with the addition of the TTC mechanism, the control frequency can still be maintained at around 20Hz, which is a highly convincing design point for resource-constrained edge robot deployments. The RAC comparison mechanism cleverly avoids the difficulties of the Reward Model—assigning absolutely high scores to consecutive actions has always been a thorny issue in reinforcement learning or post-training. The paper transforms this into a relative preference judgment of "whether A is better than B," which not only reduces the modeling difficulty but also, because of the elimination system, is naturally suitable for parallel acceleration. The data construction approach is also ingenious: directly using the number of ODE steps generated during flow-matching to define positive and negative samples saves on manual annotation and provides a practical solution to the problem of a lack of high-quality preference data in the VLA field.


weakness: there may be biases caused by the automatically constructed data distribution: the performance degradation caused by reducing the number of ODE steps is more reflected in trajectory smoothness than in logical errors at the decision-making level. Although Appendix A conducted cross-task experiments, the distinctions between those tasks were relatively obvious. I am somewhat skeptical about whether RAC can accurately identify subtle errors that lead to task failure when "the same task and the trajectory are smooth." Secondly, the threshold τ of the "cognitive clutch" is currently heuristically set based on offline data. The robustness of this static threshold is questionable in completely undistributed scenarios—if the system frequently triggers TTC due to unfamiliar environments, will real-time performance be reduced to unusable levels? Finally, the baseline comparison is not comprehensive enough: mainly comparing it to the 1.5Hz. this "generational" comparison has limited persuasiveness. If the basic PI0.5 is paired with simple Best-of-N sampling and a lightweight discriminator, how much of the VLA-ATTC lead can be maintained? More ablation experiments of intermediate states are needed to corroborate this.

---

> ### Author Rebuttal · Authors · 2026-03-29
>
> We sincerely thank the reviewer for the constructive feedback and for recognizing several strengths of our work, especially the practical amortized single-prefill/batch-decoding design for real-time deployment, the relative RAC formulation that avoids unstable absolute scoring, and the automated preference-pair construction strategy. We are grateful for these thoughtful comments. Below we clarify the raised points and report additional experiments.
>
> >### 1. RAC behavior on semantically incorrect yet smooth actions
>
> Thank you for this valuable suggestion. To directly test the case “same task, smooth trajectory, but semantically wrong action,” we added a new **real-robot diagnostic benchmark**. On **3 real-world tasks**, we manually collected failure states where the policy produced a bad action, e.g., **selecting the wrong target object** or **starting interaction before reaching the correct pose**. For each such state, we stored the corresponding **KV cache** and **proprioceptive state**, then switched to **teleoperation** to collect a human good action at the **same state**, forming tightly matched positive/negative pairs.
>
> | New evaluation set | #Tasks | #Pairs | RAC accuracy |
> |---|---:|---:|---:|
> | Real-robot smooth-but-wrong pairs | 3 | 300 | **93.0% (279/300)** |
>
> This result suggests that RAC does not merely capture trajectory smoothness; it can distinguish **good vs. bad actions within the same task and the same state**, even when the bad action remains visually smooth.
>
> >### 2. Cognitive clutch under deployment shifts
>
> We appreciate this important question regarding the robustness of the static threshold. We therefore added a **real-robot OOD experiment** on **3 tasks**, where we randomly **increase or decrease lighting intensity** in each trial to simulate a common deployment-time visual shift.
>
> | OOD setting (random lighting perturbation) | TTC trigger ratio | 3-step consecutive trigger ratio | Avg. control frequency |
> |---|---:|---:|---:|
> | Real-robot, 3 tasks | **22.4%** | **1.1%** | **20.1 Hz** |
>
> Under this perturbation, TTC activation remains stable and does **not** degenerate into frequent consecutive triggering. Task success also remains close to normal lighting: **60 / 66 / 60** under normal lighting versus **58 / 62 / 62** under randomized lighting. Finally, even in our original **full TTC** setting (i.e., triggering TTC at every step), the system still runs at **12.1 Hz**, which remains practically usable (higher than OpenVLA’s 3–5 Hz control frequency).
>
> >### 3. Additional Best-of-N baselines
>
> Thank you for suggesting stronger comparisons. We added two stronger baselines on **LIBERO-LONG**:
> (1) **BoN-16 + GPT-5.4 scorer**, which samples 16 actions and uses GPT-5.4 to score them under the current scene;
> (2) **BoN-16 + ResNet50-MLP absolute critic**, supervised by the GPT-5.4 scores.
>
> | Method | Control freq. | PI0 avg. success | PI0.5 avg. success |
> |---|---:|---:|---:|
> | Base model | 23.3Hz | 82.8 | 90.6 |
> | BoN-16 + GPT-5.4 scorer | 4.8Hz | 86.2 | 91.4 |
> | BoN-16 + ResNet50-MLP absolute critic | 21.5Hz | 83.4 | 90.2 |
> | **VLA-ATTC** | **20.8Hz** | **90.6** | **94.0** |
>
> These baselines provide some gains, but they either incur a **substantial control-frequency drop** or yield only **limited improvement**, and they remain below **VLA-ATTC**. This suggests that the improvement is not simply from sampling more actions or adding a generic discriminator, but from the combination of **adaptive clutch + relative critic**.
>
> >### 4. RAC deployment overhead
>
> Thank you for asking for clearer deployment statistics. We additionally profiled RAC as follows:
>
> | Metric | Value |
> |---|---:|
> | RAC parameters | **325M** |
> | Single pairwise comparison | **~3 ms** |
> | One TTC overhead (16 candidates + tournament) | **~70 ms** |
>
> Since base PI0 / PI0.5 inference latency is already close to **90 ms**, this overhead is moderate in practice and remains compatible with real-time deployment.
>
> >### 5. Scope and fail-safe handling
>
> We appreciate this thoughtful question. VLA-ATTC is designed for settings where the base VLA already has **non-trivial capability** in the current scenario but behaves **unreliably**; our goal is to suppress bad actions and increase the chance of selecting a good one **within its action distribution**. If the base VLA has **no viable behavior at all** in a state, this is outside the current target scope of VLA-ATTC. We will clarify this assumption more explicitly in the revision, and we agree that adding an **absolute reject / emergency-stop mechanism** under all-bad candidates is an important future direction.

---

### Decision · Program_Chairs · 2026-04-30

**Decision:**

Accept (regular)

**Comment:**

This paper proposes VLA-ATTC, a test-time method for improving the performance of VLA models. The core idea is to estimate action uncertainty and select better actions using a critic model.

**Strengths**

* Strong experimental results in both simulated and real-world settings

* Clear and well-structured writing

**Weaknesses**

* Limited baseline comparisons (e.g., TACO, best-of-N with alternative models)

* Insufficient evaluation across diverse benchmarks

Most of these concerns were addressed during the rebuttal. This work makes a solid contribution to the community, and I therefore recommend weak accept.